Studying kinematic linkage of finger joints: estimation of kinematics of distal interphalangeal joints during manipulation

Roda-Sales Alba rodaa@uji.es
Sancho-Bru Joaquín L.
Vergara Margarita
Departamento de Ingeniería Mecánica y Construcción, Universitat Jaume I , Castelló de la Plana , Castelló , Spain
Yakovenko Sergiy
Electronic publication date: 2022 Oct 4
Publication date: 2022
Volume: 10
Electronic Location ID: e14051
Received 2022 Apr 22; Accepted 2022 Aug 23
Copyright: ©2022 Roda-Sales et al.
Copyright year: 2022
Copyright holder: Roda-Sales et al.
License: This is an open access article distributed under the terms of the Creative Commons Attribution License, which permits unrestricted use, distribution, reproduction and adaptation in any medium and for any purpose provided that it is properly attributed. For attribution, the original author(s), title, publication source (PeerJ) and either DOI or URL of the article must be cited.
License URL: https://creativecommons.org/licenses/by/4.0/

Keywords: Hand, Hand kinematics, Hand joints, Interphalangeal joints, Manipulation, Kinematic linkage, Biomechanics

Funding: Ministerio de Ciencia, Innovación y Universidades (MCIU) Agencia Estatal de Investigación (AEI) European Regional Development Fund (ERDF) PGC2018-095606-B-C21 This work was supported by Ministerio de Ciencia, Innovación y Universidades (MCIU), Agencia Estatal de Investigación (AEI) and European Regional Development Fund (ERDF) through the project PGC2018-095606-B-C21. The funders had no role in study design, data collection and analysis, decision to publish, or preparation of the manuscript.

==============================
The recording of hand kinematics during product manipulation is challenging, and certain degrees of freedom such as distal interphalangeal (DIP) joints are difficult to record owing to limitations of the motion capture systems used. DIP joint kinematics could be estimated by taking advantage of its kinematic linkage with proximal interphalangeal (PIP) and metacarpophalangeal joints. This work analyses this linkage both in free motion conditions and during the performance of 26 activities of daily living. We have studied the appropriateness of different types of linear regressions (several combinations of independent variables and constant coefficients) and sets of data (free motion and manipulation data) to obtain equations to estimate DIP joints kinematics both in free motion and manipulation conditions. Errors that arise when estimating DIP joint angles assuming linear relationships using the equations obtained both from free motion data and from manipulation data are compared for each activity of daily living performed. Estimation using manipulation condition equations implies a lower mean absolute error per task (from 5.87° to 13.67°) than using the free motion ones (from 9° to 17.87°), but it fails to provide accurate estimations when passive extension of DIP joints occurs while PIP is flexed. This work provides evidence showing that estimating DIP joint angles is only recommended when studying free motion or grasps where both joints are highly flexed and when using linear relationships that consider only PIP joint angles.

Introduction

The complexity of human hand kinematics, with more than 25 main degrees of freedom, provides the ability required to perform activities of daily living (ADLs), ensuring a full and autonomous life. ADL performance requires manipulation of a wide variety of products with different shapes and design characteristics. The characterization of its kinematics has to consider the different phases involved: reaching, grasping, manipulation and object release. Nevertheless, measuring certain joints during product manipulation is challenging. This is the case of distal interphalangeal (DIP) joints, whose recording is hindered by factors such as lack of space for locating sensors, occlusions in optical systems or improper fit of the sizing of instrumented gloves (Eccarius, Bour & Scheidt, 2012). Alternatively, DIP joint angles might be estimated by taking advantage of the kinematic linkage existing between proximal interphalangeal (PIP), metacarpophalangeal (MCP) and DIP joints, which has been attributed to the tendinous system and ligaments of the fingers (Holguin et al., 1999; Landsmeer, 1963). Several studies have contributed to the exploration and quantification of this linkage, especially that between PIP and DIP joints. Table 1 summarizes the main experimental regression values between PIP and DIP joints reported in the literature. These studies were mostly limited to the analysis of the free motion corresponding to opening and closing the fist (Hahn et al., 1995; Van Zwieten et al., 2015; Kim, 2006; Mentzel et al., 2011).

Table 1 Regressions of interphalangeal joint angles obtained in literature with DIP angle (θDIP) as the dependent variable and PIP angle (θPIP) as the independent variable.

Authors	Task/fingers analysed	Participants	Motion capture system	Regressions obtained (angles in deg.)	
Hahn et al. (1995)	Opening–closing the fist / Both index fingers	17	Ultrasound marker system	Index: θDIP = 0.76⋅θPIP	
Van Zwieten et al. (2015)	Theoretical model validated with opening–closing the fist	1	Custom-made angles-video-goniometry	S-shape curves with parameters dependent on subject’s anatomy, generic for index to little fingers. Mean slope in central linear zone ≈ 0.75	
Kim (2006)	Opening–closing the fist/Right hand fingers	1	CyberGlove instrumented glove	Index: θDIP = 0.6175⋅θPIP + 0.4199
Middle: θDIP = 0.4715⋅θPIP + 0.7023
Ring: θDIP = 0.4390⋅θPIP + 0.7336
Little: θDIP = 0.4143⋅θPIP + 0.5665	
Mentzel et al. (2011)	Opening–closing the fist/Right hand fingers	10	Customized instrumented glove	Index: θDIP = 0.77⋅θPIP
Middle: θDIP = 0.75⋅θPIP
Ring: θDIP = 0.75⋅θPIP
Little: θDIP = 0.57⋅θPIP	

All the regressions presented in Table 1 assumed zero offset, except the one presented Kim (2006), where the experimental offset observed was negligible (<1°), although data were obtained from a single subject. The experimental slopes observed during free motion follow a similar distribution among fingers in all the studies, being higher for the index finger, followed by the middle, ring and little (Kim, 2006; Mentzel et al., 2011). Values for the index finger are similar in the studies with the highest number of subjects, and smaller in the study with a single subject. From this, it can be hypothesized that this type of experiment benefits from large sample sizes, as anatomical differences in recruited subjects may affect results. All these aforementioned studies analyzed the PIP-DIP linkage by performing controlled and guided free motion, but none of them considered the performance of real or simulated tasks representative of ADLs.

Several works in the literature also included MCP recording when studying the kinematic linkage of finger joints (Kim, 2006, Mentzel et al., 2011; Darling, Cole & Miller, 1994; Gülke et al., 2010; Nakamura et al., 1998). Some of them provided the coefficients for second-order curves to obtain the position of fingertips for prosthetic applications (Kim, 2006), as well as descriptive data of kinematic parameters and correlation coefficients (Mentzel et al., 2011; Darling, Cole & Miller, 1994; Gülke et al., 2010). The slopes between the index PIP and DIP joints during free motion of the index finger were observed to be much less variable than the slopes between the index MCP and PIP joints (Darling, Cole & Miller, 1994). Another study analyzed MCP, PIP and DIP flexion profiles during the grasping of cylinders with different diameters (Gülke et al., 2010), and studied parameters such as mean flexion for each finger and diameter or mean coupling ratio of the maximum flexion angle. Nevertheless, none of these studies provided any equations correlating MCP joint angles with those of PIP and DIP during task performance or explored such a possibility.

The aim of this work is, therefore, to contribute to the study of the kinematic linkage between MCP, PIP and DIP joints, not only in free motion, but also during manipulation. To do so, this work proposes the measurement of finger joint kinematics during free motion tasks and a set of ADLs representative of the most commonly performed tasks, using different products and performing different grasp types. It then aims to obtain equations to estimate DIP joint kinematics from these sets of data, taking advantage of the kinematic linkage. And finally, it will estimate joint angles using these equations, in order to quantify the error that arises when estimating DIP joint angles in manipulation.

Materials & Methods

Participants

Nine healthy adult participants volunteered to take part in the experiment, approved by the Universitat Jaume I Ethics Committee (approval reference number CD/31/2019). Minimum hand length required to be eligible for recruitment was 184 mm, so as to avoid any fitting problems presented by the instrumented gloves (oversized for small and medium hands), in accordance with the minimum hand length established in previous works (Roda-Sales, Sancho-Bru & Vergara, 2022). Therefore, all the participants were males (six right-handed and three left-handed, aged 32.7 ± 12.2 years), with a mean hand length of 192.9 mm (SD 7.8 mm). All the participants were previously informed about the characteristics of the experiment and gave their written consent.

Material

One left- and one right-hand 22-sensor CyberGlove III were used in the experiment, together with the objects required to perform the tasks (Fig. 1).

Figure 1 Scenario and objects required to perform the set of ADLs.

Experimental conditions

The joint angles of the participants were recorded with the instrumented gloves in two different experimental conditions: (i) performance of free motion tasks (FMT), and (ii) performance of tasks representative of ADLs. The order of performance of the ADLs or FMT experimental condition was randomized for each subject.

Free motion tasks

Subjects were asked to perform two free motion tasks seated in front of a table, resting their elbows on the table and maintaining their hands vertically (Fig. 2) while wearing the instrumented gloves. In the first free motion task (FMT1) subjects were asked to flex and extend the PIP and DIP joints of the four fingers three times at a moderate, self-selected pace (in order to ensure that the range of motion was fully covered), keeping the MCP joints of their fingers in a neutral position (not flexed, with proximal phalanges aligned with metacarpals, see left figures of FMT1 and FMT2 of Fig. 2) and the thumb extended (Fig. 2, left). In the second free motion task (FMT2) they were asked to flex and extend the MCP, PIP and DIP joints of the four fingers (Fig. 2, right), in accordance with the movement in previous studies (Hahn et al., 1995; Van Zwieten et al., 2015; Kim, 2006; Mentzel et al., 2011), again keeping the thumb extended.

Figure 2 Performance of FMT1 (left) and FMT2 (right).

Tasks representative of ADLs

Table 2 shows the complete list of ADLs performed in the experiment. The tasks consisted of the 20 ADLs proposed in the Sollerman Hand Function Test (SHFT) (Sollerman & Ejeskär, 1995) as being representative of the activities and grasp types performed by a healthy adult subject during daily life. Moreover, six additional ADLs were performed in order to include the grasp types under-represented in the SHFT (intermediate, special pinch and non-prehensile) according to the real frequency of grasps in ADLs (Vergara et al., 2014). All the subjects performed the tasks following the operator’s instructions, which included whether subjects had to use both hands or only the dominant one according to SHFT instructions (Sollerman & Ejeskär, 1995) (see Table 2). Subjects were asked to maintain a controlled initial and final posture in each task, with their hands lying at their sides in a relaxed position, with fingers slightly flexed. Time stamps were marked by the operator during the recordings of the ADLs when the subject started and finished the contact with the manipulated objects. In this way, the data collected were separated into three phases: (i) from initial relaxed posture to object contact (i.e., reaching), (ii) object manipulation, (iii) from end of object contact to final relaxed posture (i.e., release).

Table 2 ADLs performed in the experiment.

Marked with “x” when using both hands was allowed.

ID	Both hands	ADL	
1		Picking up a coin from flat surface, putting it into a purse mounted on a wall	
2		Opening/closing zipper	
3		Picking up a coin from a purse	
4		Lifting wooden cubes over an edge 5 cm in height	
5		Lifting an iron over an edge 5 cm in height	
6		Turning a screw with a screwdriver	
7		Picking up nuts and putting them on bolts	
8		Putting a key into a lock, turning it 90°	
9		Turning a door-handle 30°	
10	x	Tying a shoelace	
11		Unscrewing lids of jars	
12	x	Doing up buttons	
13		Putting a tubigrip stocking on the other hand	
14	x	Cutting play dough with a knife and fork	
15		Eating with a spoon	
16		Writing with a pen	
17	x	Folding a piece of paper and putting it into an envelope	
18	x	Putting a paper-clip on an envelope	
19	x	Writing with a keyboard	
20		Lifting a telephone receiver, putting it to the ear	
21	x	Pouring water from a carton	
22	x	Pouring water from a jug	
23	x	Pouring water from a cup	
24	x	Putting toothpaste on a toothbrush	
25		Spraying the table with a cleaning product	
26		Cleaning the table with a tea towel	

Data analysis

A previously validated protocol (Gracia-Ibáñez et al., 2017) was used to calculate the flexion angles at the MCP, PIP and DIP joints of fingers 2 to 5 of the right and left hands from the data recorded by both CyberGloves, acquired at a frequency of 100 Hz. The angles of the dominant hand of each subject were selected and then low-pass filtered (2nd order Butterworth filter, cut-off frequency 5 Hz), and static initial and final data of all recordings were trimmed. The recordings of tasks representative of ADLs were split into a manipulation phase (ADL_M) and reaching plus release phases (ADL_R), using the time stamps marked by the operator during the recordings, in order to distinguish free motion (ADL_R) from manipulation (ADL_M). Therefore, 28 sets of free motion data were collected for each subject (FMT1, FMT2 and ADL_R1 to ADL_R26), and 26 sets of manipulation data (ADL_M1 to ADL_M26).

In order to achieve appropriate statistical power so as not to commit type II errors (given that the analyses were planned to be performed with the data collected), and also to reduce the computing time required, each set of data (FMT1, FMT2, ADL_R1 to ADL_R26 and ADL_M1 to ADL_M26) were reduced to 10 samples each, as this sample size provided a statistical power close to 0.8. The samples were equally distributed along the task time using linear interpolation (Fig. S1 shows a set of data before and after resampling in order to illustrate that no important information is lost in one of the most manipulative tasks performed). Henceforth, unless otherwise specified, all analyses refer to these reduced sets of data throughout the text.

Regression type selection for free motion data

First, a decision was made regarding the set of data and the type of linear regression to use in order to be representative of the kinematic linkage in free motion conditions. Three sets of data were considered: FMT1, FMT2 and ADL_R. The significance of the regression coefficients obtained, the DIP range of motion covered and the mean absolute errors using different regression types were compared for each set of data in order to select the most appropriate set of data for further regression analyses.

Two aspects were studied in order to select the most appropriate type of linear regression: the independent variables and the possibility of considering null or non-null constant coefficient. To do so, for each subject, finger, and set of data, six linear regressions were performed, derived from combining a different set of independent variables and null/non-null constant coefficient. DIP flexion was always the dependent variable. The three different combinations of independent variables were: (1) only PIP flexion; (2) PIP flexion and MCP flexion; and (3) PIP, MCP and interaction of PIP and MCP flexion. Then, the statistical significance (α ≤ 0.01) of the coefficients of independent variables was checked.

Furthermore, repeated measures ANOVAs (α ≤ 0.05) were performed with DIP ranges of motion in FMT1, FMT2 and ADL_R in order to check which set of data was better at covering ranges of motion and, therefore, at providing more appropriate data to perform regressions.

Then, in order to determine the appropriateness of considering null or non-null constant coefficient, using the selected set of data and the regression type with the selected independent variables, mean coefficients across subjects (both with null and non-null constant coefficients) were obtained for each finger. After this, these mean coefficients were used to estimate the DIP joint angles in this same set of data. Mean absolute errors of these estimations across subjects were compared with repeated measures ANOVAs to determine whether considering constant coefficient was appropriate or not. Finally, a set of 4 equations (one per digit) was obtained as a proposal to estimate DIP angles from free motion data (EQ_F).

Selection of regression type for manipulation data

Again, two aspects were studied in order to select the most appropriate type of linear regression: the independent variables and the possibility of considering null or non-null constant coefficient. Regressions were performed using the ADL_M data of the 26 ADLs altogether for each subject, always with DIP flexion as the dependent variable and considering null and non-null constant coefficient and the same three combinations of independent variables explained in the previous section, and the significance (p ≤ 0.01) of the independent variable coefficients was checked.

Then, in order to determine the appropriateness of considering null or non-null constant coefficient, using the regression type with the selected independent variables, the mean coefficients across subjects (with both null and non-null constant coefficient) were obtained for each finger and were used to estimate the DIP joint angles in ADL_M. The most appropriate type of regression was selected from errors, as described in the previous section, by comparing them with repeated measures ANOVAs. Therefore, another set of four equations (one per digit) were selected as an alternative proposal to estimate the DIP angles (EQ_M), but in this case obtained from manipulation data. Figure 3 presents an overview of all the performed regressions with the different sets of data in order to determine both EQ_F and EQ_M.

Figure 3 Diagram with the process followed to determine EQ_F and EQ_M.

Joint angles estimation

Afterwards, both sets of equations (EQ_F and EQ_M) were used to estimate DIP angles during ADL_M and ADL_R phases. The differences between the estimated DIP angles and those recorded at each instant were computed. Two hundred and eight (26 tasks × 2 phases × 4 fingers) repeated measures ANOVAs with one degree of freedom were applied on these errors to check for significant differences between the set of equations used.

Results

Regression type selected for free motion

The range of motion for the DIP joint in the recordings using FMT2 data was lower than that using FMT1 data for all the fingers. The mean DIP range of motion across subjects was 48.52° vs. 67.46° for the index finger, 44.76° vs. 76.94° for the middle finger, 42.64° vs. 63.27° for the ring finger, and 71.98° vs. 75.09° for the little finger, during FMT2 and FMT1, respectively. The repeated measures ANOVAs revealed that differences were statistically significant for middle finger (p = 0.038), but not for index (p = 0.082), ring (p = 0.107) and little (p = 0.803). The DIP range of motion was impeded because of contact of the fingertips with the palm (Fig. 4), on some occasions presenting DIP joint extension. Consequently, the FMT2 task was discarded, as these extension values were considered not to be representative of fingers free motion.

Figure 4 DIP flexion limited by the contact of fingers with palm.

The regressions using ADL_R data with the MCP joint as one of the independent variables provided more than 50% of the MCP regression coefficients non-statistically significant. In contrast, when considering PIP joint flexion as an independent variable the data presented high linearity and most coefficients were statistically significant. Therefore, regression with the PIP joint angle as the only independent variable was considered the most appropriate for the subsequent analyses.

DIP ranges of motion during FMT1 were in general higher than during ADL_R, consequently providing more appropriate data to perform regressions. The mean DIP range of motion across subjects was 67.46° vs. 50.51° for the index finger, 76.94° vs. 72.59° for the middle finger, 63.27° vs. 54.99° for the ring finger, and 75.09° vs. 81.58° for the little finger, during FMT1 and ADL_R, respectively. The repeated measures ANOVAs revealed statistically significant differences for index finger (p = 0.009), but not for middle (p = 0.396), ring (p = 0.128) or little finger (p = 0.379). Therefore, given the observed tendency of higher ranges of motion and the statistically significant differences obtained in index finger, the FMT1 set of data was selected to obtain free motion coefficients.

Finally, the most appropriate regression type (null or non-null constant coefficient) was selected by comparing the error that arises when using coefficients obtained in both types of regression conditions. The mean absolute errors when estimating DIP angles from the PIP ones in FMT1 were (null vs. non-null coefficient): 6.01° vs. 6.35° for the index finger, 9.38° vs. 9.58° for the middle finger, 6.91° vs. 7.02° for the ring finger and 7.48° vs. 8.19° for the little finger. The repeated measures ANOVAs revealed statistically significant differences for middle finger (p = 0.047), but not for index (p = 0.126), ring (p = 0.630) or little (p = 0.093). Therefore, given the observed tendency of lower errors when estimating using the null constant coefficient regression type, and the statistically significant differences obtained in middle finger, the regression with the null constant coefficient was chosen for the set of FMT1 equations. Table 3 presents descriptive statistics across subjects of the regressions with the null constant coefficient performed for each finger during the FMT1, all with p ≤ 0.01.

Table 3 Descriptive statistics of the slopes and R2 values in the regressions for each finger during FMT1.

FMT1	SLOPE	R2	
FINGER	Mean	SD	Max	Min	Mean	SD	Max	Min	
Index	0.52	0.11	0.66	0.36	0.98	0.02	0.99	0.94	
Middle	0.75	0.15	0.97	0.56	0.96	0.04	0.99	0.86	
Ring	0.52	0.11	0.71	0.38	0.95	0.05	0.99	0.83	
Little	0.80	0.13	1.04	0.67	0.97	0.04	1	0.89	

Regression type selected for manipulation

The regressions performed on ADL_M data considering the MCP joint angles as one independent variable provided more than 50% of non-statistically significant coefficients. In contrast, when considering PIP joint flexion as the independent variable the data presented high linearity and most coefficients were statistically significant. Thus, the decision was again taken to select a regression type only considering PIP.

The mean absolute errors across subjects when estimating DIP joint angles in ADL_M from regressions considering only the PIP joint angle as the independent variable were (null vs. non-null constant coefficient): 8.65° vs. 8.61° for the index finger, 13.09° vs. 13.19° for the middle finger, 10.31° vs. 10.06° for the ring finger and 11.57° vs. 10.93° for the little finger. The repeated measures ANOVAs revealed statistically significant differences for ring (p = 0.003) and little finger (p = 0.000), but not for index (p = 0.107) and middle finger (p = 0.315). These errors were also computed for each task (Figs. S2 to S9). Although the error was lower in some tasks and fingers when performing estimations using null constant coefficient, the overall errors were slightly lower when performing estimations using non-null constant coefficient in most fingers except for the middle finger, where it was similar. Furthermore, almost all the constant coefficients (28 out of 36) were statistically significant (p ≤ 0.01) and so this regression type was chosen as the most appropriate one. Table 4 presents descriptive statistics across subjects of these regressions performed for each finger during the ADL_M of the 26 ADLs altogether, again all with p ≤ 0.01.

Table 4 Descriptive statistics of the slopes, constant coefficients (in degrees) and R2 values in the regressions for each finger during the ADL_M of the 26 ADLs altogether.

ADL_M	Slope	Constant coeff.	R2	
FINGER	Mean	SD	Max	Min	Mean	SD	Max	Min	Mean	SD	Max	Min	
Index	0.44	0.15	0.71	0.22	−2.47	4.76	4.76	−5.66	0.48	0.19	0.81	0.13	
Middle	0.81	0.19	1.22	0.59	−13.97	8.87	0.04	−28.31	0.65	0.14	0.87	0.35	
Ring	0.58	0.12	0.86	0.49	−12.33	7.56	−3.71	−23.98	0.63	0.10	0.77	0.44	
Little	0.87	0.20	1.21	0.65	−10.52	9.16	4.36	−21.50	0.69	0.15	0.88	0.46	

Estimated joint angles and observed errors

Scatter plots of DIP vs. PIP angles (showing all the data recorded) for each finger and phase (ADL_R and ADL_M) for each subject are presented in Figs. S10 to S17. The plots represent data recorded in the 26 ADLs (a different colour per task) and the FMT regression line for each subject and finger. Analogue scatter plots but including all the data recorded in FMT are presented in (Figs. S18 to S21). The mean absolute errors across subjects when estimating ADL_R and ADL_M data using FMT and ADL_M coefficients are presented in Table 5.

Figures S22 to S25 present the box and whiskers plots of the errors (for each finger and task) of estimating the DIP angles during ADL_R using both the coefficients obtained during FMT and during ADL_M conditions. The tasks that presented the highest mean absolute errors when performing estimations using FMT coefficients and ADL_M coefficients are presented in Table 6, along with the value of the mean absolute error across subjects.

The repeated measures ANOVAs revealed significant differences (sig. ≤ 0.01, average observed power of 0.745) in several tasks between the estimations of the DIP angles during the ADL_R phase, using FMT or ADL_M coefficients. Tables S1 and S2 present obtained p value and partial eta squared for each repeated measures ANOVA. Table 7 lists the tasks that presented the lowest error when estimating angles using the coefficients from each condition, per finger. Those that presented statistically significant differences are highlighted in bold.

Table 5 Mean absolute errors across subjects when estimating ADL_R and ADL_M data using FMT and ADL_M coefficients.

	Estimation of ADL_R	Estimation of ADL_M	
Finger	With FMT coef.	With ADL_M Coef.	With FMT coef.	With ADL_M Coef.	
Index	6.54°	4.07°	10.15°	8.61°	
Middle	12.80°	9.78°	15.65°	13.19°	
Ring	10.80°	7.69°	12.74°	10.06°	
Little	11.04°	8.28°	12.62°	10.93°	

Table 6 Tasks with highest and lowest mean absolute errors across subjects when estimating ADL_R data using FMT and ADL_M coefficients.

	With FMT coefficients	With ADL_M coefficients	
Finger	Highest mean abs. error	Lowest mean abs. error	Highest mean abs. error	Lowest mean abs. error	
Index	13. Putting a tubigrip on (9.00°)	21. Pouring water from a carton (4.47°)	2. Opening/closing a zipper (5.87°)	21. Pouring water from a carton (2.38°)	
Middle	4. Lifting wooden cubes (17.87°)	26. Cleaning the table (6.49°)	22. Pouring water from a jug (13.67°)	12. Doing up buttons (7.03°)	
Ring	2. Opening/closing a zipper (15.75°)	26. Cleaning the table (4.02°)	13. Putting a tubigrip on (11.08°)	11. Unscrewing the lid of jars (4.98°)	
Little	2. Opening/closing a zipper (15.84°)	26. Cleaning the table (6.70°)	5. Lifting an iron (10.89°)	8. Putting a key into a lock and turning it (6.00°)	

Table 7 Tasks classified depending of the mean error when estimating DIP angles from PIP ones in ADL_R, classified by fingers.

Tasks that presented statistically significant differences when applying the ANOVA are highlighted in bold.

	ADL_R	
	Tasks with the lowest error with FMT coefficients	Tasks with the lowest error with ADL_M coefficients	
Index		1, 2, 3, 4, 5, 6, 7, 8, 9, 10, 11, 12, 13, 14, 15, 16, 17, 18, 19, 20, 21, 22, 23, 24, 25, 26	
Middle	15, 16, 17, 21, 22, 23, 25, 26	1, 2, 3, 4, 5, 6, 7, 8, 9, 10, 11, 12, 13, 14, 18, 19, 20, 24	
Ring	14, 16, 17, 18, 21, 22, 23,25, 26	1, 2, 3, 4, 5, 6, 7, 8, 9, 10, 11, 12, 13, 15, 19, 20, 24	
Little	26	1, 2, 3, 4, 5, 6, 7, 8, 9, 10, 11, 12, 13, 14, 15, 16, 17, 18, 19, 20, 21, 22, 23, 24, 25	

Figures S26 to S29 present box and whiskers plots of the errors (for each finger and task) in estimating the DIP angles during the ADL_M phase using both the coefficients obtained during FMT and ADL_M conditions. The tasks that presented the highest absolute mean errors when performing estimations using FMT coefficients and ADL_M coefficients are presented in Table 8, along with the value of the mean absolute error across subjects.

Table 8 Tasks with highest and lowest mean absolute errors across subjects when estimating ADL_M data using FMT and ADL_M coefficients.

	With FMT coefficients	With ADL_M coefficients	
Finger	Highest mean abs. error	Lowest mean abs. error	Highest mean abs. error	Lowest mean abs. error	
Index	16. Writing with a pen (22.86°)	21. Pouring water from a carton (4.79°)	16. Writing with a pen (17.83°)	1. Picking up a coin (4.31°)	
Middle	11. Unscrewing the lids of jars (23.66°)	26. Cleaning the table (7.97°)	23. Pouring water from a cup (18.87°)	26. Cleaning the table (9.12°)	
Ring	4. Lifting wooden cubes (19.30°)	26. Cleaning the table (4.12°)	5. Lifting an iron (15.04°)	19. Writing with a keyboard (5.15°)	
Little	20. Lifting a telephone receiver (20.52°)	26. Cleaning the table (7.34°)	20. Lifting a telephone receiver (19.19°)	1. Picking up a coin (4.95°)	

The repeated measures ANOVAs revealed significant differences (sig. ≤ 0.01, average observed power of 0.824) in several tasks between the estimations of the DIP angles during the ADL_M phase, using FMT or ADL_M coefficients. Table 9 lists the tasks that presented the lowest error when estimating angles using the coefficients from each condition, per finger. The ones that presented statistically significant differences are highlighted in bold.

Table 9 Tasks classified depending on the mean error when estimating DIP angles from PIP ones in ADL_M, classified by fingers.

Tasks that presented statistically significant differences when applying the ANOVA are highlighted in bold.

	ADL_M	
	Tasks with the lowest error with FMT coefficients	Tasks with the lowest error with ADL_M coefficients	
Index	2, 4, 5, 9, 21, 22	1, 3, 6, 7, 8, 9, 10, 11, 12, 13, 14, 15, 16, 17, 18, 19, 20, 23, 24, 25, 26	
Middle	3, 5, 6,14, 20, 21, 22, 26	1, 2, 4, 7, 8, 9, 10, 11, 12, 13, 15, 16, 17, 18, 19, 23, 24, 25	
Ring	5, 6, 13, 21, 22, 26	1, 2, 3, 4, 7, 8, 9, 10, 11, 12, 14, 15, 16, 17, 18, 19, 20, 23, 24, 25	
Little	5, 6, 9, 13, 22, 26	1, 2, 3, 4, 7, 8, 10, 11, 12, 14, 15, 16, 17, 18, 19, 20, 21, 23, 24, 25	

Discussion

Data linearity and regression coefficients

The joint flexion linkage of fingers has been studied in free motion and manipulation during a set of representative ADLs. High linearity both in free motion and in manipulation was observed between PIP and DIP joint flexion data, and most of the correlation coefficients when performing linear regressions considering DIP flexion as the dependent variable were statistically significant. This observed linearity and correlation between PIP and DIP joint flexion is coherent with previous studies presenting coefficients between the kinematics of both joints (Hahn et al., 1995; Van Zwieten et al., 2015; Kim, 2006; Mentzel et al., 2011), and is mainly attributable to the anatomy and tendinous system of finger joints (Van Zwieten et al., 2015). Nevertheless, this significance in regression coefficients and data linearity was not observed when also considering MCP joint flexion as an independent variable. Therefore, in order to estimate DIP joint angles only PIP joint flexion was considered, both in free motion and in manipulation conditions. The appropriateness of considering constant coefficients in regressions was also studied. Regression type with non-null constant coefficient was selected as most appropriate for manipulation phase data, while regression with null constant coefficient was selected for free motion data. This is in accordance with the consideration of null or negligible constant coefficients in previous works in the literature studying PIP-DIP linkage during free motion (Hahn et al., 1995; Van Zwieten et al., 2015; Kim, 2006; Mentzel et al., 2011; Gülke et al., 2010).

In contrast to many studies in the literature, this work considered analyzing the PIP-DIP linkage in free motion using three different sets of data: the reaching phase of tasks, the task of closing the fist and a task flexing PIP and DIP, but maintaining the MCP joint in a neutral position. This comparison went a step further than other experiments in the literature that only analyze the task of closing the fist (Hahn et al., 1995; Van Zwieten et al., 2015; Kim, 2006; Mentzel et al., 2011), thereby helping us to determine the free motion dataset providing the best fitting regressions.

The data selected to perform regressions with PIP flexion as the independent variable and DIP flexion as the dependent one presented high linearity, both in free motion and in manipulation conditions. The slopes obtained in free motion conditions are within the range of values reported in the literature (Table 1). However, they are larger for the middle and little fingers (0.75 and 0.80, respectively) than for the index and ring fingers (0.52), and this distribution of slopes among fingers does not match the ones reported in the literature, which are not consistent either. These differences may be attributable to the way of performing the free movement in the experiments. While other works considered a movement of closing the fist (Hahn et al., 1995; Van Zwieten et al., 2015; Kim, 2006; Mentzel et al., 2011), in FMT1 participants were asked to keep the MCP joints in a neutral position while PIP and DIP joints were flexed, so as to separate the PIP-DIP flexion relationship from the MCP flexion. Moreover, the movement of closing the fist, used in the reported works, could have limited DIP flexion on some occasions because of the contact of the fingertips with the palm, as happened in FMT2 (Fig. 4), which is not exactly representative of pure free motion. Nevertheless, the aim of several of these works (Hahn et al., 1995; Van Zwieten et al., 2015) was analysing PIP-DIP flexion relationship in order to discriminate healthy from pathological fingers, rather than to estimate joint kinematics.

The slopes obtained herein could have been affected to a lesser extent by the stiffness of the instrumented glove. Nevertheless, this stiffness is expected to affect both PIP and DIP flexion to a similar extent, thus not affecting the flexion ratio significantly.

Mean slopes across subjects obtained for middle, ring and little fingers are higher in manipulation conditions than in free motion (0.81 vs. 0.75 in the middle finger, 0.58 vs. 0.52 in the ring finger and 0.87 vs. 0.80 in the little finger). Nevertheless, they are balanced out in manipulation by significant offsets of −13.97° (middle finger), −12.33° (ring finger) and −10.52° (little finger). The index finger is the only one that presents a lower slope in manipulation than in free motion (0.44 vs. 0.52). Furthermore, it presents the lowest R squared value (0.48) among all the fingers and phases when performing the regression with manipulation data. This lower slope and bad fit may be attributable to simultaneous active PIP flexion and passive DIP extension occurring during certain grasp types, especially pinch grasps (see Fig. 5), because the kinematic chain collapses when external forces are applied on the distal phalanx, therefore becoming negative slope values. This can be clearly observed in the scatter plots of PIP vs. DIP of the index finger during manipulation (Fig. S14). This passive DIP extension during PIP flexion, apart from reducing the mean slope values for this finger, also becomes a worse data fit.

Figure 5 Grasp with active flexion of the index PIP joint and passive extension of the index DIP joint.

The scatter plots of DIP vs. PIP angles during the reaching phase of tasks (Figs. S10 to S13) demonstrate that the PIP-DIP linkage in the free motion during ADLs (i.e., ADL_R) is quite similar to that of the free motion task (except in some ADLs). Despite the fact that, in general terms, the data fit the linear regression obtained during the free motion task quite well, the range of motion is lower in the reaching phase and in some specific tasks the PIP joint flexes while the DIP joint is kept in an almost neutral position. This happens only in some subjects, probably owing to their specific ligamentous system: when approaching an object to perform certain grasps (e.g., a 2- or 3-finger pinch), the fingers that do not participate in the grasp are folded away by flexing the PIP joints while the DIP joints remain in a neutral position (Fig. 6, left). The DIP joints can be passively extended in other cases when the fingertips come into contact with the palm (Fig. 6, right).

Figure 6 Posture of middle to little fingers during reaching.

(Left) Middle to little fingers (which do not participate in the grasp) folded away during reaching. (Right) Middle to little fingers (which do not participate in the grasp) with passive DIP extension during reaching.

The scatter plots for both the reaching phase and the free motion task (Figs. S10 to S18, respectively) show a linear relationship for the index finger. Nevertheless, data from the middle, ring and little fingers of certain subjects seem to fit better to a parabolic function (Figs. S11 to S13 and Figs. S19 to S21, as the DIP joints do not experience any flexion for low PIP flexion.

In contrast, scatter plots of DIP vs. PIP angles during manipulation show poor linearity (Figs. S14 to S17), and only in a few fingers and subjects do the data fit approximately to the corresponding free motion regression line. The index finger is the one with the most extreme data points (i.e., farthest from the regression line), as it is generally more involved in grasping than the other fingers. These extreme data points are usually under the free motion regression line, but rarely above it. Again, this is due to the passive DIP extension or to maintaining a neutral posture during PIP flexion. This configuration is largely more common during manipulation than flexing the DIP joint while the PIP is kept neutral (which would generate points above the free motion regression line). This is unnatural even during manipulation (note the reference to the PIP neutral position, rather than extension, as this joint has almost no extension range of motion).

Estimation of DIP joint angles in manipulation phase

The error that arises when estimating the DIP angle from the PIP angle in manipulation data using manipulation and free motion coefficients were presented in Table 9. It can be observed that those tasks that present the lowest errors when estimated using free motion coefficients are the ones that require a cylindrical grasp for their performance, and the diameter of the object to be grasped is small. Among these tasks, those that present the statistically significant lowest error in more than one finger are lifting an iron (#5), pouring water from a jug (#22) and cleaning the table (#26).

In contrast, those that present the lowest errors when estimated using manipulation coefficients are those that require a grasp where passive extension of the DIP joint can appear while flexing the PIP joint (such as pinch or non-prehensile grasps) as a consequence of the pressure applied during the grasp, and also because of the shape of the object being manipulated. Furthermore, as mentioned previously, when performing certain grasps (e.g., a 2- or 3-finger pinch), some subjects tend to fold away the fingers that do not participate in the grasp by flexing the PIP joints while keeping the DIP joints in a neutral position. These tasks that presented the statistically significant lowest error in more than one finger are putting a coin into a purse (#1), zipping/unzipping a purse (#2), picking up a coin from a purse (#3), lifting wooden cubes (#4), putting nuts on bolts (#7), putting a key into a lock (#8), tying a shoelace (#10), unscrewing lids off of jars (#11), doing up buttons (#12), eating with a spoon (#15), writing with a pen (#16), folding a piece of paper and putting it into an envelope (#17), putting a paperclip on an envelope (#18), writing with a keyboard (#19), pouring water from a cup (#23), putting toothpaste on a toothbrush (#24) and spraying the table with a cleaning product (#25).

Estimation of DIP joint angles in reaching phase

As regards the error that arose when estimating the DIP angle in the reaching phase using manipulation and free motion coefficients, Table 7 clearly shows that only the task of cleaning the table with a tea towel (#26) presents the statistically significant lowest errors in more than one finger when performing the estimation using free motion coefficients. In contrast, many tasks present the statistically significant lowest error in more than one finger when estimated using manipulation coefficients: putting a coin into a purse (#1), zipping/unzipping a purse (#2), picking up a coin from a purse (#3), lifting wooden cubes (#4), lifting an iron (#5), using a screwdriver (#6), putting nuts on bolts (#7), putting a key into a lock (#8), turning a door-handle (#9), tying a shoelace (#10), unscrewing the lids off of jars (#11), doing up buttons (#12) and putting a tubigrip on (#13). This is attributable to the fact that the PIP and DIP joints do not achieve the same degree of flexion in the reaching phase as in the free motion task (FMT1) (see scatter plots for ADL_R and FMT). As mentioned previously, data in the reaching phase presents a parabolic fitting shape, as the DIP does not start to flex until a certain degree of PIP flexion is achieved. Therefore, the regression line of the reaching phase data would be more similar to that of manipulation (lower slopes) than to that of the free motion task.

Comparison of observed errors using free motion and manipulation coefficients

Box and whiskers plots of the errors that arise when estimating data present a higher dispersion in the manipulation phase (Figs. S22 to S25) than in the reaching phase of the tasks (Figs. S26 to S29), but all of them present a similar bias. It is remarkable that for all the phases, fingers and tasks, differences between measured and estimated DIP joint angles are larger when estimated using free motion coefficients than when using the manipulation ones. Therefore, free motion coefficients tend to overestimate the DIP flexion angles: even though manipulation slopes are higher than free motion ones (except for the index finger), the negative constant coefficients in manipulation regressions significantly reduce the estimated flexion values. Even though the estimation using manipulation condition coefficients implies a lower mean absolute error per task (see Table 10), it fails to provide accurate estimations when passive extension of DIP joints occurs while PIP is flexed, as postures are quite dependent on the shapes of the objects and the pressure applied during grasping.

The magnitude of the obtained errors when using both types of coefficients could be acceptable in several applications such as virtual reality used in rehabilitation, or teleoperation, among others. Nevertheless, joint and tendon forces may be significantly affected by these postural errors, as moment arms would be affected by these changes in posture. Therefore, it may have an important impact in different applications, such as in biomechanical analyses in research or when planning surgical interventions like tendon transfers.

Table 10 Maximum mean absolute error per task when using both types of coefficients.

	Mean absolute error per task with ADL_M coefficients	Mean absolute error per task with FMT coefficients	
Index	<5.87°	<9°	
Middle	<13.67°	<17.87°	
Ring	<11.08°	<15.75°	
Little	<10.89°	<15.84°	

Conclusions

The main outcome of this work has been the assessment of the error that arises when estimating DIP joint angles assuming an experimental linear relationship with the PIP joint angles, depending on the task performed (and, consequently, on the grasp type used). The estimation of the DIP joint angles using the slopes obtained from free motion conditions implies low absolute errors in grasps or tasks where both PIP and DIP are highly flexed. Even though the estimation using manipulation condition coefficients implies a lower mean absolute error per task (from 5.87° to 13.67°) than using the free motion ones (from 9° to 17.87°), it fails to provide accurate estimations in many cases: passive extension of DIP joints may occur while PIP is flexed, and postures are quite dependent on the shapes of the objects and the pressure applied during grasping. Therefore, in view of the results from this study, estimating DIP joint angles from PIP ones and taking advantage of their kinematic linkage is only recommended if studying free motion or grasps where both joints are highly flexed and using free motion coefficients, but not in other conditions. The mean error under these conditions, taking the tasks that presented statistically significant lower errors for each finger, was 5.92° for the index finger, 12.21° for the middle, 8.61° for the ring and 11.12° for the little.

Nevertheless, this work presents some limitations, such as the stiffness of instrumented gloves, which may affect the resultant motion and therefore, results of this work. Future works could consider the anatomical variability of the sample participants to achieve better estimations, in particular by considering the range of DIP extension.

Supplemental Information

Supplemental Information 1 Supplemental Figures and Tables

Click here for additional data file.

The authors would like to thank all the study participants for their collaboration.

Additional Information and Declarations

Competing Interests

Author Contributions

Human Ethics

Data Availability

The authors declare there are no competing interests.

Alba Roda-Sales performed the experiments, analyzed the data, prepared figures and/or tables, authored or reviewed drafts of the article, and approved the final draft.

Joaquín L. Sancho-Bru conceived and designed the experiments, authored or reviewed drafts of the article, and approved the final draft.

Margarita Vergara conceived and designed the experiments, authored or reviewed drafts of the article, and approved the final draft.

The following information was supplied relating to ethical approvals ({i.e.}, approving body and any reference numbers):

Universitat Jaume I Ethics Committee.

The following information was supplied regarding data availability:

The experiment data is available at Zenodo: Alba Roda-Sales, Joaquín L. Sancho-Bru, & Margarita Vergara. (2022). Studying kinematic linkage of finger joints: Experiment data [Data set]. Zenodo. https://doi.org/10.5281/zenodo.6451662.

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
