# Peer review of "Studying kinematic linkage of finger joints: estimation of kinematics of distal interphalangeal joints during manipulation"

_PeerJ, doi:10.7717/peerj.14051_

## Round 0.1 · original submission · Major Revisions

There is a general consensus that this submission has good quality. The reviewers need mostly clarifications. The revision should focus on the questions related to the experimental design and the validity and f results.

Reviewer 1 ·

Basic reporting

• Define acronyms the first time a term is used, and once an acronym is introduced, ensure it is used consistently. For example, distal interphalangeal joint is used on line 19, but the acronym is not defined until Line 20. Additionally, DIP is spelled out on Line 26.
• The motion capture dataset, which includes activities of daily living and free hand motions, would be a valuable contribution to the scientific community, if the authors chose to make it public.

Experimental design

• On Line 107, the authors state the first free motion task was completed “at a normal pace.” Can the authors be more specific, in case speed is important to anyone attempting to replicate these results?
• On Line 124, the authors state that the starting and ending postures is “a natural position.” Can the authors provide a qualitative description (e.g., slightly flexed finger? Straight fingers?) to enhance reproducibility?
• On Line 145, the authors indicate that data was reduced to 10 samples. How did the authors ensure that no important information was lost from the signal when reducing data sampled at 100 Hz over several seconds to 10 samples? Including a supplemental figure illustrating raw data and downsampled data for a representative task would be useful.
• On Line 158-159, the notation makes it challenging to follow the independent variables. Did the authors consider the interaction of PIP and MCP flexion? Is that what the dot between PIP and MCP representing?
• A figure summarizing all of the tested regressions would enhance clarity of Section 2.5 and 2.6.

Validity of the findings

• The authors should be commended for collecting a large dataset that includes a wide variety of free motions and activities of daily living. However, it appears that the final analyses only use a subset of these data for the presented regression equations. As written, it is unclear why the regression could not be performed on all data? What is the advantage to only performing the regression on the subset of data with the largest range of motion?
• The authors acknowledge that some data shows poor linearity and even may be better fit by a parabolic function (Lines 378-387). Additionally, the reported errors are quite high (upwards of 20 degrees for some tasks). These findings seem to imply that linear regression is not appropriate for estimating DIP kinematics. Yet, the authors conclude (Line 316-326) that the assumption of linearity is valid and the regression approach is useful. The Conclusion presents a more nuanced interpretation of the Results. The Discussion would be easier to follow if the nuanced interpretation presented in the Conclusion and supported by the Results was introduced earlier.

Additional comments

• The Abstract could be strengthened by including more methodological detail. As written, it is hard to understand from the Abstract alone how the coefficients were defined and to what extent all data versus subsets of data were used in analyses.
• On Line 86, the authors state the minimum hand length is established based on prior work. It would be helpful to cite this work.
• Lines 265-280 and 291-306 present ordered lists in text. These data would be easier to read as an ordered list in a Table. Then the text could simply provide a summary of key findings (e.g., magnitude and ranges of error across tasks).
• The Discussion is challenging to follow. The authors make a few interesting observations regarding the anatomical function of the hand ligaments and/or how hand kinematics vary by task type. However, these observations get lost in the myriad of details presented on the regression equations. Consider dividing the Discussion into sections to make contributions to scientific understanding of the hand versus specific understanding of the role of regression equations easier to follow.

Reviewer 2 ·

Basic reporting

No comment

Experimental design

No comment

Validity of the findings

No comment

Additional comments

First comment - One of the main conclusions, that interdependent DIP-PIP angles which show linear relationships, are best seen in free and unresisted movements of the fingers, is welcomed very much to be underlined again, however, it is quite evident from functional anatomy ! Namely, as soon as resisting external forces are applied to a freely flexing finger, the stability of its kinematic chain which normally shows an beautifully arched form, may change - the so-called "collapsing of the kinematic chain".
Second comment - In Table 1, Motion Capture System of Reference [5] is left blank. This is not fair, because the authors clearly indicate on their p. 131 of this Reference [5] : "Data were directly derived
from angles, measured on video stills of healthy and ulnar-minus fingers in this patient, acquired under
standard conditions by the use of techniques and methodologies recently described by Hahn et al."
Moreover, these measurements were double-blinded (by a colleague from the acknowledgements).
So, instead of leaving this space blank, it should better be filled out like 'Custom-made angles-video-goniometry', or such. The same method was applied in small animals, e.g. Varejão 2002, Narain 2013.
Because here no gloves with sensors were applied, nor in Hahn's 1995 method (nor in Holguín's 1999), this can also be responsible for differences between the various outcome data results. We 'll try to attach one page from Faridi Narain's PhD thesis with a short description and an example of this method.
Third and last comment - with much appreciation we follow all finely measured differences in steepness of the graphs with those found in literature. But don't forget, that initially such graphs were intended to discriminate between healthy and pathological fingers, as clearly indicated by Hahn in a last sentence at his page 699. His suggestion to measure intrinsic minus fingers too ('ulnar nerve paralysis') became finally true in Reference [5].
Short and good - in addition to this admirable manuscript, it will be appreciated if the above remarks will be taken into account somewhere in the manuscript to show that the authors know the backgrounds too

Annotated reviews are not available for download in order to protect the identity of reviewers who chose to remain anonymous.

Reviewer 3 ·

Basic reporting

In the manuscript the English used is clear most of the time. However, long sentences (longer than 4 lines or more) are present in many places which make reading and comprehension a bit challenging, especially when many factors, such as tasks, data sets, equations, and digits, are involved in this study. For example, in lines 175-178, it is not clear how the regression for ADL_M data were performed for each digit. Was it performed on each ADL item or they are lumped together for regression? Similar issues are found in other places in the manuscript as well.

Line 124: The ‘neutral Position’ should be clearly defined..

Lines 142-145: The method employed for reducing data acquired from each task to 10 data points should be describe clearly.

Lines 217-227: To assess whether to include a constant coefficient in the final equation to predict DIP angle, the authors compared magnitudes of absolute error (absolute value of difference between measured and predicted DIP angles) derived from equations with or without constant coefficients. The authors reported absolute errors for both conditions and made a decision to use the equation with a smaller absolute error. However, differences in values of absolute error were very small between these two equations. Were the differences statistically significant? If yes, provide the statistical results. If not, they are practically the same. Therefore, choice between them becomes arbitrary. Same issue appears in Lines 238-245 for the ADL_M tasks.

Long sentences with too much detailed numerical data packed into it are present in the Conclusion.

Supp. Figures 17-20 cited in line 265 do not match the statement. Do you mean Suppl. Figures 21-24??

Supp. Figures 21-24 cited in Line 291 do not match the statement. Do you mean Suppl. Figures 25-28??

Suppl. Figures 21-28. It would help the readers to understand the figures if more detailed info are provided. For example, what do stars and circles represent??

A discussion of the limitations of the present study may be necessary.

Experimental design

The one of the main issues of this research is to reconfirm the DIP-PIP kinematic linkage in free finger movements as well as during manipulation. The selection of manipulation tasks (SHFT and others) are based on their importance in hand function and not necessarily based on their being representatives of categories of different kinematic patterns. The authors may want to sort these 26 ADL tasks into several kinematic categories (for example, types of grips or their combinations) for easier further analyses latter and may facilitate the understanding of the readers.

The criteria for selecting the data set for further regression analyses (such as magnitude of DIP joint ROM, significance of regression coefficients, absolute error) should be clearly stated in the method.
When DIP extension is present in manipulation tasks it may warrant a separate set of equation for use to derive the DIP angle from that of PIP. As DIP extension tends to enhance flexion of the PIP, therefore, leading to a negative slop for the equation, a greater variability on slop, a smaller slop average, and a larger absolute error for DIP estimate. A separate equation for subjects with digits exhibiting DIP extension (let’s call them ‘extensors’ for sake of clarity) during manipulation not only helps in more accurately estimating DIP angle in these ‘extensors’, but also leads to a better equation with a much more accurate estimates of DIP angle for the ‘flexors’ population.

Another issue along the same line of thinking is the need for separate equations for ‘contact digit(s)’ (may include those with indirect contact) and ‘non-contact digit(s)’ for manipulation tasks by using EQ_M for ‘contact digits’ and EQ_F for ‘non-contact digit(s)’, respectively.

Validity of the findings

An important issue in this manuscript is that authors jettisoned a better set of equations (EQ_M) with lower absolute error according to the prescribed criteria in favor of the other set of equations (EQ_F) even though the latter failed to accurately predict instances when DIP extension occurred during manipulation. As was suggested previously, this subgroup of subjects (digits) may warrant a separate set of equations for a better fit and reduce the absolute errors estimated both in DIP extension cases as well as DIP flexion cases.

Lines 196-201: The authors eliminated FMT2 data from further analyses on the bases that the DIP angle measured based on FMT2 data was smaller than that derived from FMT1 as DIP flexion was impeded by fingertips contacting the palm and in some cases even caused DIP joints to rest in extended position. It would be more convincing to the reader if evidence is provided such that PIP angle derived from FMT2 is a worse regressor in predicting DIP angle than that derived from FMT1.

Lines 215-216. DIP range of motion was greater in FMT1 than that of ADL_R. Please provide statistical results with p values to be certain that DIP range of motion was greater in FMT1.
Again, DIP range of motion was the sole criterion for eliminating ADL_R data for further selection for the regression equation. It would be more convincing to the reader if evidence is provided such that DIP angle derived from ADL_R data is a worse regressor in predicting DIP angle than that derived from FMT1.

Additional comments

Lines 238-242: The differences in the values of absolute errors are small between DIP angles estimated from equations derived from ADL_M with non-null and null constant. Are such differences statistically significant?

Lines 293-316: Irrespective of the outcome of the statistical analyses, are magnitudes of such absolute errors acceptable for applications in research or clinical practice.

---

## Round 0.2 · Minor Revisions

Thank you for the thorough revision that addressed all the main concerns. Please consider polishing the minor remaining issues identified by the reviewers. These are minor modifications of text that will improve clarity.

Reviewer 1 ·

Basic reporting

No comment

Experimental design

- I appreciate the authors' argument that speed does not influence their analyses. But, it may influence how others use the published datasets. Therefore, I would recommend explicitly stating that it was a self-selected pace, as this tells the reader that speed was not controlled. (Suggested edit: "three times at a moderate, self-selected pace").
- The addition of Figure 3 is great. The flowchart adds a lot of clarity.

Validity of the findings

No comment

Additional comments

No comment

Reviewer 2 ·

Basic reporting

No comment

Experimental design

No comment

Validity of the findings

No comment

Additional comments

I have read all authors' responses on my remarks as a "Reviewer No. 2", in the Rebuttal Letter. I also found back these responses in the form of adjustments, extra sentences and remarks, etc. in their corrected manuscript. This is greatly appreciated, because in this way, more clarity is created.

Still, their promised sentence "According to the comment, the space in the table referring to the “Motion capture System” of reference [5] has been filled now with “'Custom-made angles-video-goniometry”." is still absent in Table 1. I sincerely hope, and I presume, that this sentence still shows up in the final text !!!

By doing so, the authors will also prove that they have read their references well - moreover, that they leave the original authors of this specific reference in their rights ! Therefore, I fully trust that they will eventually add their promised extra sentence in their final text.

Reviewer 3 ·

Basic reporting

The authors have addressed all my previous comments in a satisfactory fashion. I have only a few minor comments.

Line 29. No results were presented in the abstract. Add a sentence or two to describe the most important results before the conclusion.

Experimental design

Lines 112: In this case, a definition of the ‘zero position’ where the angular readings of MCP, PIP and DIP joints are zero, such as left figures of FMT1 and FMT2 of Figure 2, may be necessary.

Lines 147-150: This sentence is long and a bit awkward. The authors may consider deleting ‘the data for’ in line 149. The authors might want to further elaborate or justify the use of 10 (and not, for example, 20) samples (after resampling) for regression analyses from the statistical and practical points of view and cite references if available.

Line 169 and line 171. Suggest replacing line 169 ‘p ≤ .01’ and line 171 ‘p ≤ .05’ to ‘α = .01’ and ‘α = .05’, respectively.

Validity of the findings

None

Additional comments

Line 236-237. ‘…. but not for middle (p = 0.396), ring (p = 0.128) or little (p = 0.379)’. Suggest revising ‘little (p = 0.379).’ to ‘little finger (p = 0.379).’

Lines 249-251. This sentence should be in the discussion section and not in the results section.

Line 275. ‘(Sig ≤ 0.01)’ should be ‘(p ≤ 0.01)’.

The revised discussion section is now a lot easier to follow. However, it still includes some excessively long sentences. For example, lines 466-472 contain a single sentence of 7 lines. Detailed numbers can be referred to a table, if possible. This sentence also leads with a most likely misplaced word ‘nevertheless’.
I would suggest the manuscript be edited by someone native to the English language.

---

## Round 0.3 · accepted · Accept

Dear authors,

Thank you for the appropriate responses to the reviewers. The manuscript is ready for publication. I have noticed that there are still some grammatical errors throughout the text. For example, "... it fails to provide accurate estimations when passive extension of DIP joints occur while PIP ..." should be "... occurs ...". I suggest correcting these typos during the proofing stage with a careful revision focused on spelling and grammar. Thank you for the high-quality submission and for publishing your work in PeerJ.

With regards,
Sergiy